# Development and Validation of a Cyber-Physical System Leveraging EFDPN for Enhanced WSN-IoT Network Security

**DOI:** 10.3390/s23229294

**Published:** 2023-11-20

**Authors:** Sundaramoorthy Krishnasamy, Mutlaq B. Alotaibi, Lolwah I. Alehaideb, Qaisar Abbas

**Affiliations:** 1Department of Information Technology, Jerusalem College of Engineering (Autonomous) Pallikaranai, Chennai 600100, Tamil Nadu, India; 2College of Computer and Information Sciences, Imam Mohammad Ibn Saud Islamic University (IMSIU), Riyadh 11432, Saudi Arabia; motaibi@imamu.edu.sa (M.B.A.);

**Keywords:** wireless sensor network (WSN), internet of things (IoT), security, cyber-physical system, intrusion detection, farmland fertility feature selection (F^3^S), deep perceptron network (DPN), tunicate swarm optimization (TSO)

## Abstract

In the current digital era, Wireless Sensor Networks (WSNs) and the Internet of Things (IoT) are evolving, transforming human experiences by creating an interconnected environment. However, ensuring the security of WSN-IoT networks remains a significant hurdle, as existing security models are plagued with issues like prolonged training durations and complex classification processes. In this study, a robust cyber-physical system based on the Emphatic Farmland Fertility Integrated Deep Perceptron Network (EFDPN) is proposed to enhance the security of WSN-IoT. This initiative introduces the Farmland Fertility Feature Selection (F^3^S) technique to alleviate the computational complexity of identifying and classifying attacks. Additionally, this research leverages the Deep Perceptron Network (DPN) classification algorithm for accurate intrusion classification, achieving impressive performance metrics. In the classification phase, the Tunicate Swarm Optimization (TSO) model is employed to improve the sigmoid transformation function, thereby enhancing prediction accuracy. This study demonstrates the development of an EFDPN-based system designed to safeguard WSN-IoT networks. It showcases how the DPN classification technique, in conjunction with the TSO model, significantly improves classification performance. In this research, we employed well-known cyber-attack datasets to validate its effectiveness, revealing its superiority over traditional intrusion detection methods, particularly in achieving higher F1-score values. The incorporation of the F3S algorithm plays a pivotal role in this framework by eliminating irrelevant features, leading to enhanced prediction accuracy for the classifier, marking a substantial stride in fortifying WSN-IoT network security. This research presents a promising approach to enhancing the security and resilience of interconnected cyber-physical systems in the evolving landscape of WSN-IoT networks.

## 1. Introduction

A wireless sensor network (WSN) [1,2], which is made up of different kinds of sensors with limited resources, is an important part of monitoring an environment and sending important data to a designated node, also called a sink, through different communication protocols. These data are then relayed to a base station for meticulous analysis and processing, catered to the specific demands of contemporary applications. Renowned for their efficacy in remote monitoring, WSNs have a promising future, finding applicability in critical domains such as border surveillance, industrial inspection, commercial utilities, health monitoring, and environmental and infrastructure surveillance [3].

Conversely, the Internet of Things (IoT) [4,5] embodies an intricate network of interconnected smart devices tasked with the collection, processing, optimization, and dissemination of valuable data through internet channels. Each device, identifiable by a unique IP address or identifier, facilitates autonomous data exchange, enhancing the convenience and efficiency of daily activities through technological advancements [6,7,8]. However, this burgeoning development is not devoid of challenges, predominantly concerning security [9].

The extensive integration of IoT into daily life and the surge in remote device operations necessitate a unified platform facilitating seamless communication amongst a diverse array of devices [10,11,12]. This prerequisite has spurred the creation of specific IoT frameworks, outlining the architectural blueprint for selected applications and thus working towards standardizing IoT security protocols.

WSN and IoT [13,14] stand as potent forces capable of spearheading a societal transformation towards a smarter, more connected world. Despite their distinctive characteristics, they are occasionally utilized interchangeably owing to similarities in their processing power, memory storage, and communication capabilities. Both networks hold remarkable potential in real-time applications [15,16,17], yet they suffer from persistent security challenges at the device level [18].

In this context, the adoption of lightweight, low-power security mechanisms is crucial, aiming to enhance network longevity by minimizing power consumption during the intrusion detection phase [19,20,21]. Traditional methods have endeavored to address security concerns using minimal power consumption strategies, albeit with varying degrees of success. As the IoT landscape expands, so does its vulnerability to external threats, making the development of robust security infrastructure imperative.

Current classifiers have trouble differentiating normal and unusual system behaviors because there are so many network traffic data, which have different features [22]. The presence of irrelevant features and network communication disturbances further exacerbate this challenge, leading to increased resource expenditure and diminished detection rates. Therefore, the nuanced selection of features has emerged as a vital aspect of machine learning designed to accurately encapsulate object properties while eliminating redundant data [2].

Existing research has extensively explored the potential of feature selection and machine learning algorithms in network security and traffic monitoring. Nevertheless, conventional intrusion detection approaches exhibit significant limitations, including increased computation time, reduced reliability, and heightened complexity. So, the goal of this study is to develop a new Intrusion Detection System (IDS) framework to protect WSN-IoT networks as shown in Figure 1. This was achieved by using new feature optimization and classification strategies to improve accuracy and detection rates while cutting down on time [23].

### 1.1. Research Contribution

The proposed Emphatic Farmland Fertility Integrated Deep Perceptron Network (EFDPN) aims to bolster WSN-IoT network security through an innovative cyber-physical system. This method involves a systematic workflow encompassing cyber-dataset collection, preprocessing, and feature extraction, followed by implementing the Farmland Fertility Feature Selection (F3S) technique and Deep Perceptron Network (DPN) classification. In the final step, Tunicate Swarm Optimization (TSO) is employed to refine the sigmoid transformation function, enhancing prediction accuracy in intrusion detection. The major contributions of this work are as follows:(1)An Emphatic Farmland Fertility Integrated Deep Perceptron Network (EFDPN)-based cyber-physical system was developed for protecting WSN-IoT networks;(2)By using the Farmland Fertility Feature Selection (F3S) algorithm, the processes of incursion identification and classification are streamlined, with reduced computing complexity;(3)A Deep Perceptron Network (DPN) classification technique was used to accurately classify intrusion types, yielding great performance outcomes;(4)A Tunicate Swarm Optimization (TSO) model was used to estimate the sigmoid transformation function for better classification;(5)Using well-known cyber-attack datasets, the results of the proposed EFDPN model were validated and contrasted.

### 1.2. Paper Organization

The further portions of this paper are split into the following sections: Section 2 reviews the literature relevant to cyber-security and intrusion detection in WSN-IoT networks, along with their merits and demerits. Section 3 presents the overall explanation for the proposed EFDPN model with system workflow and stage-wise descriptions. Section 4 validates and contrasts the results of the proposed EFDPB-based security framework using several performance measures. Section 5 provides a summary of the entire paper along with the conclusions, results, and suggested next steps.

## 2. Related Works

This section investigates various intrusion detection approaches used to safeguard WSN-IoT networks, wherein the positives and negatives of each model are discussed based on their performance.

Pundir et al. [24] investigated the different types of security challenges in WSN-IoT networks. The different types of security requirements were also discussed in this study for protecting WSN-IoT networks from intrusions. The following categories of potential threats could greatly affect WSN-IoT networks: eavesdropping, impersonation attacks, DoS attacks, malware attacks, database attacks, and man-in-the-middle attacks. Baraneetharan et al. [25] discussed the impacts of using machine learning algorithms for intrusion detection in WSN-IoT systems. In this study, classification, regression, and clustering-based machine learning algorithms were discussed with regard to intrusion detection in WSN-IoT networks. Moreover, the suggested intrusion detection approaches were compared based on the parameters of prediction accuracy, memory requirements, network architecture, and energy consumption. Among other models, the hybrid IDS frameworks are more suitable for WSN due to their improved energy efficiency and precise detection operation. Jiang et al. [26] deployed a lightweight Gradient Boost Mechanism (GBM)-based cyber-physical system for smart-networking environments. Amouri et al. [27] designed a cross-layered IDS-framework-based linear regression model for increasing the security of WSN-IoT networks. The authors aim to detect common malicious activities like blackholes, flooding, and DDoS within networks [28]. The suggested model has the major drawbacks of an increased false-positive rate and time consumption for attack detection.

Singh et al. [29] presented a comprehensive review to examine the different types of machine-learning-based intrusion detection approaches. This paper covers a few well-known and recently developed ML algorithms to highlight their strengths and weaknesses. This will assist researchers in choosing the best algorithm for their studies. Damasevicius et al. [30] utilized a new annotated dataset named LITNET-2020 for classifying normal and intrusive events pertaining to WSN-IoT systems. In addition, the authors suggested some other cyber-attack datasets for IoT security. Safaldin et al. [31] implemented a binary gray-wolf optimization algorithm incorporated with the standard SVM mechanism for detecting intrusions in WSNs. When recommending a fitness function for assessing each subset of the selected feature, the significance of accuracy and the overall number of features were taken into account. According to the total number of features, the prediction performance of the classifier was determined in the cited work. Here, the SVM uses a dimensionality-reduced feature set for intrusion identification and classification. Some of the merits of using SVM include better scalability, high process speed, and low complexity with a reduced feature set.

Krishnan et al. [32] introduced an anomalous intrusion detection and prevention protocol for WSN-IoT networks. The authors aimed to increase the reliability of a network and provide an expanded time frame for an organization. Jayanayudu et al. [33] utilized hybrid Shuffled Frog Leap (SFL) and Ant Lion Optimization (ALO) algorithms to develop an intrusion detection framework for protecting WSN-IoT systems. Typically, securing data while improving energy efficiency is one of the most challenging network problems in present times. Increased attention to security is necessary while monitoring IDS using IoT-WSN systems. The authors of the suggested paper presented a safe routing intrusion prevention architecture for IoT-WSN networks. Moreover, they concentrated on the enhancement of network efficiency and defense against fraudulent attacks. Here, the greedy strategy was used for data routing, offering energy efficient solutions with security. Hussain et al. [34] presented a comprehensive literature review examining various routing strategies for low-powered IoT systems. Here, the assessment was carried out based on identification, screening, eligibility, and inclusion. Moreover, their work investigated the strengths and limitations of several security-based routing methodologies used in WSN-IoT networks. Al Sawafi et al. [35] implemented a hybrid deep-learning-based intrusion detection framework for WSN-IoT networks. In this paper, the authors intended to mitigate security attacks by analyzing a network traffic dataset. According to the pre-trained features, the authors’ framework categorizes normal and malicious networking traffic in the network. Maheswari and Karthika [36] constructed a multi-tiered intrusion detection (MDIT) framework for safeguarding WSN-IoT networks. Here, the Spotted Hyena Optimization (SHO) algorithm, integrated with the standard LSTM deep learning algorithm, was used to detect malicious events in cyber-data. Table 1 summarizes the limitations of state-of-the-art systems.

The following are some of the major drawbacks of the current approaches that were identified in the literature review [37,38]:Low performance in various models;Excessive memory use during classification;High curse of dimensionality;Inability to handle massive datasets.

In order to create a highly effective cyber-physical system for WSN-IoT networks, in the proposed work, we make use of innovative optimization and classification approaches.

## 3. Proposed Methodology

The proposed strategy describes how to build a cyber-physical system based on the EFDPN in order to improve the security of WSN-IoT networks. Our intention in this initiative is to adeptly identify security breaches in these networks by leveraging state-of-the-art feature selection and classification algorithms. Commencing with the acquisition of pertinent cyber datasets, the methodology transitions into a preprocessing and feature extraction phase where data are refined and pivotal features are isolated to facilitate effective intrusion detection. This is followed by the application of the Farmland Fertility Feature Selection (F3S) technique, a pivotal process designed to alleviate computational complexity by homing in on critical features. Subsequently, the Deep Perceptron Network (DPN) takes the helm, functioning as a vital tool in the precise categorization of data points and thereby playing an instrumental role in the meticulous identification of intrusions. This structured approach culminates in the integration of the Tunicate Swarm Optimization (TSO) model, fine-tuning the sigmoid transformation function in the classification phase to potentially elevate prediction accuracy. Consequently, this holistic methodology envisages a fortified security landscape for WSN-IoT networks, with a particular emphasis on enhancing the accuracy and efficiency of intrusion detection systems.

In this section of the paper, a cyber-physical system based on Emphatic Farmland Fertility Integrated Deep Perceptron Network (EFDPN) designed as a way of protecting WSN-IoT networks is described in detail. The EFDPN is utilized to enhance security in WSN-IoT environments by integrating advanced machine learning techniques, optimizing computational efficiency, reducing false positives, and demonstrating readiness for real-world applications. It offers accurate intrusion detection and quantitative performance evaluations, making it a valuable asset in safeguarding interconnected cyber-physical systems. With the aid of cutting-edge feature selection and classification algorithms, in the proposed work, we created an efficient security framework for WSN-IoT systems. The proposed EFDPN system’s workflow model is depicted in Figure 2, which consists of the following operations:Cyber-dataset collection;Preprocessing and feature extraction;Farmland Fertility Feature Selection (F3S);Deep Perceptron Network (DPN) classification;Tunicate Swarm Optimization (TSO) for sigmoid transformation function estimation.

In the proposed EFDPN framework, the emerging public intrusion detection datasets are acquired at the beginning. The next step is to conduct dataset normalization and feature extraction in order to extract the appropriate features from the given dataset [32]. The recently introduced F3S algorithm is used to select the best features by lowering dimensionality after the set of features has been extracted. This algorithm is designed to produce accurate classification results with minimal time and computational overhead. By using the features that are carefully chosen from the dataset, the DPN classifier can predict malicious events. During this process, the TSO model is utilized to optimally compute the sigmoid transfer function, which enhances the classifier’s performance in attack detection.

### 3.1. Farmland Fertility Feature Selection (F^3^S)

In this stage, a number of features are selected from the original feature set with the use of F^3^S algorithm. Feature selection or optimization is the most crucial operation in the intrusion detection system since the classifier’s detection performance is highly dependent on the features used for training and testing. A summary of earlier studies on IDS reveals that the technique of integrating predictive classifiers plays a crucial role in IDS. In contrast, the large set of data in this detection system decreases the precision as well as speed of classifiers. Hence, meta-heuristic techniques are increasingly being employed by researchers to minimize the features of data. In order to detect network assaults, the hybridization of the classifiers and the subsequent choice of useful features are essential. Here, a successful strategy for choosing the features based on F^3^S is introduced, which substantially lowers the dimensionality of features with improved accuracy. In this algorithm, the soil quality of each portion of the farm can vary from that of the others because farmers typically split various parts of a farm into distinct soil types. The quality of the soil in each segment can be changed by adding specific compounds. Therefore, farmers apply specific materials that the soil requires in order to maximize the area of each segment of the farm. Farmers alter each area of their farm in accordance with this model and by monitoring each sector’s state of the soil. Then, they may determine the feasible ways of enhancing each portion’s soil. After that, the most effective and essential materials are then distributed to each sector in order to enhance the quality of the soil. The advantages of the F^3^S algorithm are its low processing time, high convergence, and ability to reach the best solution in the searching space with minimal iterations. Initially, the feature set $F_{s}$ is obtained as the input, and the estimated $\delta_{f}$ is delivered as the output of this algorithm. After obtaining the original set of features, the number of solutions is estimated for each portion of a farm, as shown below:(1)$$H=\left(\beta \times y\right)$$
where β indicates a constant variable, and y is an integer number. After defining the initial population, the quality of each portion of the farm is determined using the following model:(2)$$Q_{\mathrm{section}}=F_{s}\left(\alpha \times d\right),\;\alpha =y\times \left(r-1\right)\;:\left(y\times r\right)\;r=\left\{1,2,\ldots ,\beta \right\},\;k=\left\{1,2,\ldots ,4\right\}$$

The average of each part is independently estimated using the aforementioned equation, where d with the interval $\left[1,\;\ldots ,\;D\right]$ is determined using the variable $F_{s}$. Then, the fitness of quality is estimated for each section as shown below:(3)$${\mathrm{fit}}_{\mathrm{section}}=\mathrm{avg}\left(\mathrm{objective}\left(F_{s}^{\mathrm{ik}}\right){\mathrm{in}\;\mathrm{section}}_{r}\right)\;r=\left\{1,2,\ldots ,\beta \right\},\;i=\left\{1,2,\ldots ,y\right\}$$
where $\mathrm{objective}\left(.\right)$ indicates the objective function, and $\mathrm{avg}\left(.\right)$ represents the average of the solutions within each section of land. Consequently, local $L^{\mathrm{mem}}$ and global memory $G^{\mathrm{mem}}$ updation are performed, and the best solution obtained from each portion is maintained in the local memory as represented below:(4)$$G^{\mathrm{mem}}=\mathrm{round}\left(t\times H\right);\;0.1<t<1$$
(5)$$L^{\mathrm{mem}}=\mathrm{round}\left(t\times y\right);\;0.1<t<1$$

Moreover, the soil quality of each portion is changed and determined using the solutions of global memory in the farm’s worst section as represented in the following models:(6)$$\rho =\tau \times \mathrm{rand}\left(-1,1\right)$$
(7)$$F_{s}^{\mathrm{new}}=\rho \times \left[F_{s}^{\mathrm{ik}}-F_{s}^{G}\right]+c$$
where $F_{s}^{G}\;$ randomly selects one of the global memory solutions, and τ is a random number between 0 and 1 that is initiated at the beginning of the algorithm. Furthermore, the solutions based on both local and global memory are updated, providing the feasible solutions in each portion, but they are not integrated with the local memory. However, some of the solutions are integrated with the best solution for improving quality, as illustrated below:(8)$$\delta_{f}=\left\{\begin{array}{c} F_{s}^{\mathrm{new}}=F_{s}^{\mathrm{ik}}+\varphi_{1}\times \left[F_{s}^{\mathrm{id}}-G^{\mathrm{best}}\right]\;\underline{R}>\mathrm{rand}\\ F_{s}^{\mathrm{new}}=F_{s}^{\mathrm{ik}}+\mathrm{rand}\left(0,1\right)\times \left[F_{s}^{\mathrm{id}}-G^{\mathrm{best}}\right]\;\mathrm{else} \end{array}\right.$$
where $\underline{R}$ indicates a random number ranging from o to 1 that represents the extent to which the solutions are combined with best global, and $\varphi_{1}$ is an integer determined at the beginning of optimization. Finally, the optimized feature set $\delta_{f}$ is obtained as the output of this algorithm, which is further used by the classifier for intrusion detection and classification. The description of F^3^S technique is presented in Algorithm 1.



**Algorithm 1: Farmland Fertility Feature Selection (F^3^S)**
Input: Feature set FsOutput: Selected Features δfStep 1: → H for each section of land, as shown in Equation (1);Step 2: → Initialize the populations, and determine the soil quality Qsection of each portion of the farm using Equation (2);Step 3: → Compute the fitness of quality solution in each portion fitsection using Equation (3); Step 4: → Perform local (Lmem) and global (Gmem) memory updation, where the best solutions in each portion are stored in the local memory using Equations (4) and (5); Step 5: → Change the quality of soil in each portion of the farm, which is determined with global memory solutions in the farm’s worst section, as shown in Equations (6) and (7); Step 6: → Update the solutions based on local memory and lobal memory Lbest, providing the feasible solutions in each section. Step 7: → Improve the quality of solutions, as depicted in Equation (8), for obtaining the optimized set of features δf.


### 3.2. Deep Perceptron Network (DPN)

After choosing the features, the DPN classifier model is applied to classify the malicious activities in a network according to their pertinent features. This is a deep learning model developed based on the multilayer perceptron neural network. The structure of the DPN is shown in Figure 3; it comprises more than three layers, including input and output layers. In this model, the network is first constructed with the number of hidden layers, and the output vector of the layer (i.e., feature map) is estimated, as shown in the following model
(9)Fml=fϑl=fWl×Fml−1+biasl
where ϑl∈ℝNl is the activation vector of the lth layer with Nl neurons, Wl ∈ℝNl*Nl−1 is the weight matrix, and biasl∈ℝNl represents the bias vector. Consequently, the sigmoid transfer function is estimated, as shown in the following model:(10)fϑ=11+e−ϑ

Typically, the activation function must be properly selected according to the type of prediction application. In the proposed framework, the activation function is optimally computed using the TSO algorithm. The posterior probability of class $j\;\in \left\{1,\ldots ,∁\right\}$ was determined to be $\mathrm{Pr}({\mathrm{cls}}_{j}|y)$. Here, the softmax function is used to satisfy the posterior probability function, as represented below:(11)$$F_{m}^{L}=\mathrm{Pr}\left({\mathrm{cls}}_{j}|y\right)=\mathrm{softmax}\left(\vartheta^{L}\right)=\frac{e^{\vartheta_{j}^{L}}}{\sum_{k=1}^{∁}e^{\vartheta_{k}^{L}}}$$
where $\vartheta_{j}^{L}$ is the element with $j^{\mathrm{th}}$ index in the activation of vector $\vartheta^{L}$. Moreover, the training process is carried out with the optimized cost function, as represented in the following equation:(12)$$Q\left(W^{l},{\mathrm{bias}}^{l}\right)=\frac{1}{S}\sum_{a=1}^{S}Q_{\mathrm{CE}}\left(W^{l},{\mathrm{bias}}^{l},y_{a},l_{f}\right)+\beta {\left|W^{l}\right|}_{F}^{2}$$

Finally, the predicted classified label can be produced, as shown in the following form:(13)$$Q_{\mathrm{CE}}\left(W^{l},{\mathrm{bias}}^{l},y_{a}\right)=-\sum_{k=1}^{C}y_{a}\left[F_{m}^{l}\right]$$
(14)$${\mathrm{cls}}_{f}=Q_{\mathrm{CE}}\left(W^{l},{\mathrm{bias}}^{l},l_{f}\right)$$

Based on this prediction operation, normal and attacking events are precisely classified in the proposed framework. Moreover, the steps to develop DPN architecture is described in Algorithm 2.



**Algorithm 2: Deep Perceptron Network (DPN)**
Input: Selected Features δf, Label data lf;Output: Classified output clsf;Procedure:Step 1: Compute the feature map Fm using Equation (9);Step 2: → Estimate the Sigmoid transformation as an activation function fϑ, as shown in Equation (10);//Tunicate Swarm Optimization;Step 3: Compute the posterior probability of class,j ∈1,…,∁ as Pr(clsj|y);Step 4: → The softmax function is used to satisfy the posterior probability normalization requirement FmL as shown in Equation (11);Step 5: → The training process is carried out with the optimized cost function as represented in Equation (12);Step 6: → The classified output is predicted as shown in the form of Equations (13) and (14);


### 3.3. Tunicate Swarm Optimization (TSO)

During classification, the sigmoid transfer function is optimally computed by using the TSO algorithm, which helps to improve the intrusion detection rate of the classifier. In general, tunicates produce a bright, pale, blue-green bioluminescent light that can be seen from a few meters away. When they reach a size of a few millimeters, these cylindrical creatures have to crack at one of their ends. Each tunicate is made up of a developing gelatinous tunic that helps to bind all the organisms together. These tunicates can grow up to a few millimeters in length and only have an opening at one of their ends. Every tunicate develops a gelatinous tunic that aids in the unification of all the individuals. By drawing water from the sea surrounding them, each tunicate uses an atrial syphon to produce jet propulsions from its aperture. A tunicate needs to meet three requirements in order to satisfy the operations of jet propulsion using the statistical model: they need to avoid collisions between possible solutions, to move further in the direction of the best solution, and to stay close to the best solution. In this technique, the feature map result $F_{m}$ is obtained as the input, and the optimal value $\vec{\varphi_{r}}$ is produced as the output. In the beginning, the parameters such as the constant $(\overset{˘}{F)}$, gravity force ($\overset{˘}{G}$), water flow advection in the deep ocean ($\overset{˘}{w_{f}}$), social force $\overset{˘}{M_{I}}$, and the maximum number of iterations are initialized as shown below:(15)$$\overset{˘}{F}=\frac{\overset{˘}{G}}{\overset{˘}{M_{I}}}$$
(16)$$\overset{˘}{G}=r_{2}+r_{3}-\overset{˘}{w_{f}}$$
(17)$$\overset{˘}{w_{f}}=2\times r_{1}$$
(18)$$\overset{˘}{M_{I}}=\left[\rho_{\mathrm{mn}}+r_{1}\times \rho_{\mathrm{mx}}-\rho_{\mathrm{mn}}\right]$$
where $r_{1}$, $r_{2}$, and $r_{3}$ are random numbers in the range [0, 1], and $\rho_{\mathrm{mn}}$ and $\rho_{\mathrm{mx}}$ are considered to equal 1 and 4, respectively. After successfully avoiding a dispute with their neighbors, the search agents move towards the best neighbors, as represented below:(19)$${ƥ}_{d}=|F_{m}-\mathrm{rand}\times \vec{\varphi_{r}}\left(x\right)|$$
where ${ƥ}_{d}\;$ is the total distance between the search agent and food source, rand is a random number in the range [0, 1], x indicates the current iteration, $F_{m}$ indicates the position of the food source, and $\vec{\varphi_{r}}$ is the position of the tunicates. The search agent may establish itself as the top search agent, as represented below:(20)$$\vec{\varphi_{r}}\left(x\right)=\left\{\begin{array}{c} F_{m}+\overset{˘}{F}\times {ƥ}_{d}\;,\;\mathrm{if}\;\mathrm{rand}\geq 0.5\\ F_{m}-\overset{˘}{F}\times {ƥ}_{d}\;,\;\mathrm{if}\;\mathrm{rand}\geq 0.5 \end{array}\right.$$

Moreover, the position of all tunicates can be updated according to the position of the first two tunicates, as shown in the following model:(21)$$\vec{\varphi_{r}}\left(x+1\right)=\frac{\vec{\varphi_{r}}\left(x\right)+\vec{\varphi_{r}}\left(x+1\right)}{2+r_{1}}$$
where $\vec{\varphi_{r}}\left(x+1\right)$ represents the updated position of the tunicates. Overall steps of TSO technique is described in Algorithm 3.



**Algorithm 3: Tunicate Swarm Algorithm (TSO)**
Input: Feature map result Fm;Output: Optimal Value φr→;Procedure:Step 1: The parameters F˘(constant), gravity force (G˘), water flow advection in the deep ocean (wf˘), social force MI˘, and the maximum number of iterations are initialized as represented in Equation (15) to (18);Step 2: → After successfully avoiding a dispute with their neighbors, the search agents are directed towards the best neighbors, as shown in Equation (19);Step 3: → The search agent can even establish its position as the leading search agent φr→x, as shown in Equation (20);Step 4: → Update the position of all tunicates in accordance with the position of first two tunicates φr→x+1 using Equation (21);Step 5: → Obtain the optimal value φr→ as the output;


## 4. Experimental Results

### 4.1. Experimental Setup

The effectiveness of the EEDPN was experimentally assessed, as shown in this section. This section contains detailed descriptions of the evaluation dataset, the experimental settings, and the experimental methods, as well as comparisons to traditional approaches, imbalanced data-processing algorithms, and cutting-edge intrusion detection techniques. To ensure efficient data handling and model training, hardware with the following specifications was used. An Intel Core i7 8th Gen processor (or higher) with a clock speed of at least 3.5 GHz was used. An NVIDIA GeForce RTX 2080 Ti GPU with 11 GB of GDDR6 VRAM was used due to its excellent deep-learning performance. The system has 16 GB of DDR4 RAM (2400 MHz) and a high-speed SSD with a 500 GB storage capacity for storing datasets, code, and model checkpoints. To support the specified hardware tools, the following software was used. The system operates on Windows 10 (64-bit). Keras 2.13 with TensorFlow as a backend for deep learning model development and MATLAB R2023b with the Statistics and Machine Learning Toolbox for machine learning algorithms were installed. Python 3.7 was used for data preprocessing and analysis. NumPy (v1.18.5) was used for numerical computations, pandas (v1.0.5) was used for data manipulation, scikit-learn (v0.23.1) was used for machine learning tasks, Matplotlib (v3.2.2) was used for data visualization, and Jupyter Notebook (v6.0.3) was used as the integrated development environment (IDE) for coding and experimentation.

At this stage, emerging benchmarking datasets like UNSW-NB 15 (intrusion dataset IS-1) and NSL-KDD (intrusion dataset IS-2) were taken into consideration for testing and validating this system. The Australian Centre for Cyber Security (ACCS) produced the UNSW-NB15 dataset in 2015. It comprises a wide range of deep-structure network communication data as well as minimal incursion information. As a result, it is better suited to imitating the complicated modern network environment. It depicts the modern network traffic mode. It has one unique attack category designation and 47 attributes. There are 2,540,044 samples in the collection, representing nine different attack methods, including fuzzers, DoS, analysis, reconnaissance, exploitation, shellcode, worm, backdoor, and generic. The NSL-KDD and UNSW-NB15 datasets, which exhibit high feature dimensions along with substantial data volume, are typical examples of high-dimensional imbalanced datasets. The majority of the data contained in them are typical network data, with only a trace quantity of attack data. Lower detection accuracy and longer training and detection times are precipitated by duplicated features and unbalanced data. The test set encompasses unidentified attacks, which puts the capacity for generalization under greater strain. Descriptions of the datasets are given in Table 2.

The NSL-KDD Dataset requires approximately 2 GB of storage space, while the UNSW-NB15 Dataset, being larger, requires around 4 GB of storage space. During deep-learning model training, the GPU memory usage ranges from 6 GB to 10 GB. This is to accommodate the model’s architecture and batch size. The NVIDIA GeForce RTX 2080 Ti provides ample memory for efficient training. The bulk of CPU usage primarily occurs during data preprocessing, where multiple CPU cores are used for parallel data processing. The extent of CPU utilization varies but typically stays below 50%. RAM usage during model training depends on the batch size and model complexity. With a batch size of 32–64, the RAM usage remains within the available 16 GB, ensuring smooth model training. The training time for the EFDPN model using the specified hardware ranges from several hours to a day. This is related to the dataset’s size, model complexity, and the number of training epochs. Adequate storage space (500 GB) is available for storing datasets, code, model checkpoints, and experiment results. Multiple experiments were conducted to assess the impact of hyperparameters and configurations. Computational overhead is incurred for each experiment. Data preprocessing, including cleaning, normalization, and feature engineering, primarily utilizes CPU resources and requires a few hours for completion, depending on the dataset’s size.

### 4.2. Performance Metrics

The standard parameters such as accuracy, precision, recall, f1-score, and training time were computed in this study in order to validate the proposed EFDPN model. These parameters determine the overall performance of the security framework, which is estimated using the following models:(22)$$\mathrm{Accuracy}=\frac{T_{+\mathrm{ve}}+T_{-\mathrm{ve}}}{T_{+\mathrm{ve}}+T_{-\mathrm{ve}}+F_{+\mathrm{ve}}+F_{-\mathrm{ve}}}$$
(23)$$\mathrm{Precision}=\frac{T_{+\mathrm{ve}}}{T_{+\mathrm{ve}}+F_{+\mathrm{ve}}}$$
(24)$$\mathrm{Recall}\;\mathrm{or}\;\mathrm{TPR}=\frac{T_{+\mathrm{ve}}}{T_{+\mathrm{ve}}+F_{+\mathrm{ve}}}$$
(25)$$F1-\mathrm{score}=2\times \left(\frac{\mathrm{Precision}\times \mathrm{Recall}}{\mathrm{Precision}+\mathrm{Recall}}\right)$$
(26)$$\mathrm{FPR}=\frac{F_{+\mathrm{ve}}}{F_{+\mathrm{ve}}+T_{-\mathrm{ve}}}$$
(27)$$\mathrm{FNR}=\frac{F_{-\mathrm{ve}}}{T_{+\mathrm{ve}}+F_{-\mathrm{ve}}}$$
where $T_{+\mathrm{ve}}$ indicates a true positive, $T_{-\mathrm{ve}}$ represents a true negative, $F_{+\mathrm{ve}}$ is a false positive, and $F_{-\mathrm{ve}}$ is a false negative. Figure 4 evaluates the performance of the proposed EFDPN model using IS-1 with and without the F^3^S mechanism.

### 4.3. Results Analysis

This section validates the performance and results of the proposed EFDPN cyber-physical system using a variety of measures and public datasets. The sample network environment created in this analysis is shown in Figure 5, where the green-colored nodes are considered normal, and other colors indicate the different types of intrusions. By using the combination of F^3^S + DPN + TSO models, intrusions can be accurately predicted in the proposed framework. For the testing and validation of this system, emerging benchmarking datasets such as UNSW-NB 15 (Intrusion dataset IS-1) and NSL-KDD (Intrusion dataset IS-2) were considered.

Similarly, the results were estimated for IS-2, as shown in Figure 6. This analysis was mainly carried out to determine the importance of using the F3S mechanism in the intrusion detection approach. The results reveal that performance was greatly improved with the use of the F3S algorithm for both datasets. As the increased dimensionality of features may affect the performance of a classifier with low accuracy, it is essential to squeeze the feature dimensionality for improved detection results. Figure 7, Figure 8, Figure 9 and Figure 10 show the results of the validation and comparison of the accuracy, precision, recall, and f1-score parameters for both conventional [39] and the proposed intrusion detection approaches with respect to the different types of attacks. These results include the following attack categories: normal attack, brute force attack, botnet attack, and web attack. A subset of the model’s performance was used to evaluate the accuracy of the algorithm. One of the metrics used to evaluate the classification models was accuracy, as computed in Equation (22). Precision implies a high rate of accurate estimation. It is a percentage of all genuine positives that the model claims are connected with all positives that the model expects, and it is estimated using Equation (23). Recall is also referred to as the true positive rate (as computed in Equation (24)), which compares the total positives in all the system states to the actual total of positives in the data. Additionally, model performance can be estimated using the F1 score, which is the weighted average of precision and recall, as computed in Equation (25). The obtained results reveal that the proposed EFDPN-based security model provides an effective attack detection result when compared to the other techniques. Due to the inclusion of the TSO and F3O algorithms, the security performance results are greatly enhanced in the proposed cyber-physical system.

Figure 11 validates the F1-score values of all the existing intrusion detection approaches in accordance with the different types of attacks. The estimated results indicate that the proposed EFDPN model triumphs over the conventional DBN-, SVM-, RNN-, SNN-, and FNN-based intrusion detection approaches, presenting an increased F1-score value. The F3S algorithm plays a vital role in obtaining better prediction results in the proposed framework since it eliminates irrelevant features in order to improve the prediction process of the classifier.

Figure 12 and Figure 13 show the results of the validation and comparison of the existing [40] and proposed security methodologies performed using the IS-1 dataset. The obtained results also indicate that the proposed EFDPN model outperforms the other models, yielding high performance results. The deployed cyber-physical system represents a more advanced and intelligent approach to intrusion detection, integrating cutting-edge techniques and optimizing the classification process. It offers improved adaptability, efficiency, and performance compared to traditional IDSs.

Figure 14 compares the conventional PSO-based classification [41] and the proposed EFDPN models using the IS-1 dataset. In order to demonstrate the effectiveness of optimization in the security framework, the optimization-integrated classifier models are compared in this study. For this assessment, standard machine learning algorithms, including RF, DT, KNN, and RC, are considered, which predict intrusions in the dataset according to the features chosen via PSO. In contrast to these algorithms, the proposed EFDPN algorithm yields improved detection results. Since the F3S technique provides the best solution with an increased convergence rate, it effectively reduces the dimensionality of features before the training and testing processes. Therefore, the proposed F3S incorporated with the DPN model greatly outperforms the other classification approaches. A comparative analysis based on the classifier’s training time was performed, as shown in Figure 15. A good classification model should minimize the training time and offer an increased attack detection rate. Typically, the classifier’s training time can be increased with the use of high-dimensionality features, which also increases the computational complexity of classification. Therefore, the input feature set used for the classifier’s training must be optimized for better predictions. According to the results, the proposed EFDPN model could effectively reduce the training time with the use of the F3S algorithm.

Figure 16 compares the accuracy of various machine learning and deep learning techniques [42] used for intrusion detection. In addition, the classifier’s detection accuracy is estimated and compared for both IS-1 and IS-2, as shown in Figure 17 and Figure 18, respectively. In addition, a qualitative security analysis [31] was also performed in this study, as shown in Table 3, based on the following parameters: false acceptance rate, accuracy, detection rate, number of features, and time consumption. Overall, the estimated results demonstrate that the proposed EFDPN cyber-physical system provides effective prediction results when compared with the existing algorithms due to the inclusion of the F3S and TSO algorithms. This is because effective dimensionality reduction was achieved with the use of the F3S technique and the classifier sigmoid function was computed with the use of the TSO algorithm.

## 5. Discussion

In this study, we embarked on a rigorous exploration of potential enhancements in intrusion detection mechanisms within the framework of WSN-IoT networks through the development and evaluation of an Emphatic Farmland Fertility Integrated Deep Perceptron Network (EFDPN)-based cyber-physical system. Our investigative journey was grounded in a substantial body of previous studies, which mapped out the present landscape of intrusion detection systems, along with their respective merits and challenges ([24,25,29,30,35,36,37,38]). The proposed EFDPN model represents a significant stride towards the fortification of WSN-IoT networks, primarily anchored by its innovative Farmland Fertility Feature Selection (F3S) mechanism and a potent classification stage leveraging a Deep Perceptron Network (DPN) followed by fine-tuning with Tunicate Swarm Optimization (TSO) for sigmoid transformation function estimation. This innovative concoction of methodologies not only hones the accuracy of intrusion detection but also astutely manages feature dimensionality, thereby mitigating computational complexity and enhancing the efficiency of the system.

When juxtaposed against existing models documented in previous studies, such as DBN-, SVM-, RNN-, SNN-, and FNN-based approaches, our model exhibits a significant escalation in performance metrics such as accuracy, precision, recall, and F1-score, as evidenced by the results derived from utilizing the benchmark datasets UNSW-NB 15 and NSL-KDD ([39,40,41,42]). Notably, the implementation of the F3S algorithm resulted in being a crucial factor in boosting the predictive efficacy of the classifier via allowing for the meticulous filtering of irrelevant features, thereby facilitating an improved prediction process and a commendable reduction in training time. However, it is imperative to acknowledge potential limitations that might encumber the proposed framework. Future studies might focus on further optimizing the computational efficiency of the EFDPN model alongside exploring its applicability and performance across diverse, more complex network environments. Additionally, a deeper dive into addressing potential vulnerabilities to newer, sophisticated attack vectors would be a prudent avenue to tread.

While the results are promising, we recognize the need for continuous evolution in optimizing computational efficiency and in tailoring the framework to counter newer, sophisticated attack vectors. Future research trajectories should also explore the scalability of the EFDPN in real-time environments with diverse infrastructures to fully realize its robustness and adaptability.

Furthermore, the scalability of the proposed model should be tested in real-time scenarios, spanning across diverse infrastructures and varying scales, to rigorously assess its robustness and adaptability. Parallelly, fostering collaborations with industry stakeholders could foster the refinement of the model to meet specific, real-world requirements and standards. In conclusion, our study stands as a testament to the viable advancements in securing WSN-IoT networks through intelligent, data-driven mechanisms. The EFDPN model, with its innovative blend of feature selection and classification methodologies, marks a promising precedent in the realm of cyber-physical systems security. As we venture forth, it holds immense potential to spearhead a new generation of resilient, efficient, and intelligent intrusion detection systems, fostering a safer and more secure cyber-physical landscape.

### 5.1. Advantages of EFDPN Model

Our study offers a novel strategy for protecting WSN-IoT networks using clever, data-driven approaches. The EFDPN model, with its novel convergence of feature selection and classification strategies, heralds a promising frontier in the security of cyber-physical systems, promising a robust, effective, and intelligent infrastructure capable of fending off the constantly evolving cyberthreats and fostering a safer and more secure cyber-physical landscape. The following are the primary advantages of the proposed system:(1)The model can precisely identify different types of incursions by combining F3S and DPN, reducing false positives;(2)The F3S method makes it easier to extract pertinent information, improving the model’s capacity to pinpoint threats with greater accuracy while requiring less computational effort;(3)By including tunicate swarm optimization (TSO), the sigmoid transformation function can be adjusted, improving the model’s ability to detect intrusions;(4)Thanks to better feature selection and decreased dimensionality, the EFDPN model efficiently decreases training time, boosting efficiency without compromising the detection rate;(5)The architecture of the EFDPN model allows for scalable deployment, making it adaptable to various network sizes and complexities;(6)The model is capable of identifying and mitigating a wide range of attack categories, including brute force, botnet, and web attacks, thereby providing a robust defense mechanism;(7)The model’s compatibility with established benchmark datasets (UNSW-NB 15 and NSL-KDD) showcases its readiness for real-world applications and further testing;(8)Given its feature set and capabilities, the EFDPN model has substantial potential for implementation in real-time environments, offering a timely response to security breaches;(9)The model is designed to minimize the usage of resources, such as memory, through intelligent design choices in the classification and feature selection phases, which contribute to overall system efficiency.

### 5.2. Future Works

The proposed EFDPN model heralds a promising frontier in the security landscape of WSN-IoT networks. In the future, the following trajectories can be pursued to further its potential:(1)Conduct pilot studies to assess the model’s adaptability and performance in real-time environments, with a focus on scaling the model to accommodate larger and more complex network infrastructures;(2)Further refine the F3S and TSO algorithms to enhance computational efficiency and accuracy, possibly integrating it with other optimization techniques to forge a more robust system;(3)Continually update and adapt the model to identify and counteract emerging and sophisticated attack vectors, fostering a dynamic security framework that evolves with the threat landscape;(4)Develop multi-layered security protocols within the EFDPN framework, which can work in synergy with existing security infrastructures, to provide a comprehensive security solution;(5)Explore the potential applications of the EFDPN model in other domains, such as industrial control systems and healthcare networks, tailoring the model to meet the unique security requirements of these sectors;(6)Engage with the user and broader community to gather feedback and insights, fostering a collaborative approach to further refine and enhance the model;(7)Develop educational initiatives and training programs to foster awareness and skill development, equipping individuals and organizations with the tools to effectively deploy and manage EFDPN-based security systems.

By pursuing these trajectories, we envision the EFDPN model evolving into a cornerstone of cybersecurity in WSN-IoT networks, setting a new standard in resilience, efficiency, and intelligence in the face of escalating cyber threats.

## 6. Conclusions

This paper introduces novel EFDPN-based cyber-physical systems designed to increase the security of WSN-IoT systems. In this study, the combination of F3S, DPN, and TSO mechanisms was implemented to construct a computationally effective and accurate intrusion detection framework. The emerging public intrusion detection datasets IS-1 and IS-2 were obtained first for processing. To extract the necessary features from the given dataset, dataset normalization and feature extraction processes were carried out. After the set of features was retrieved, the new F3S algorithm was utilized to choose the best features by reducing dimensionality. The objective of this technique is to generate precise categorization results with little computational overhead. The DPN classifier can then forecast malicious occurrences using the attributes that were carefully selected from the dataset. In this instance, the sigmoid transfer function is optimally computed using the TSO model, which improves the classifier’s effectiveness in attack detection. Moreover, standard performance measures such as accuracy, precision, recall, f1-score, and training time were estimated and compared during evaluation to demonstrate the effectiveness of the EFDPN model. Then, recent state-of-the-art models were compared with the EFDPN mechanism using IS-1 and IS-2. Overall, the obtained results reveal that the EFDPN model provides improved prediction performance over other algorithms following the inclusion of F3S and TSO algorithms. In the future, the current security framework will be enhanced to protect IoMT or IoHT from network intrusions with low complexity.

## Figures and Tables

**Figure 1 sensors-23-09294-f001:**
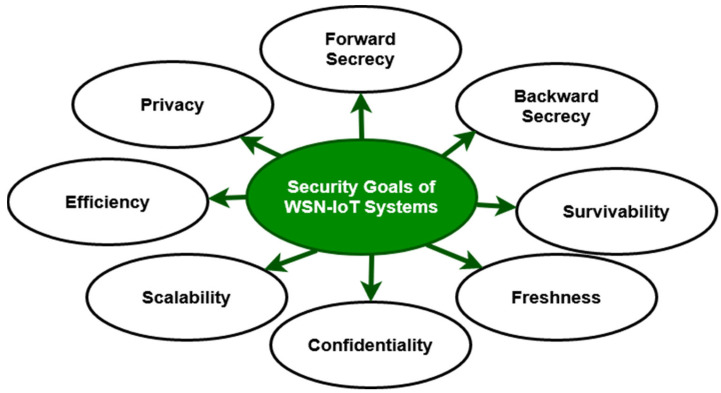
Security properties of WSN-IoT systems.

**Figure 2 sensors-23-09294-f002:**
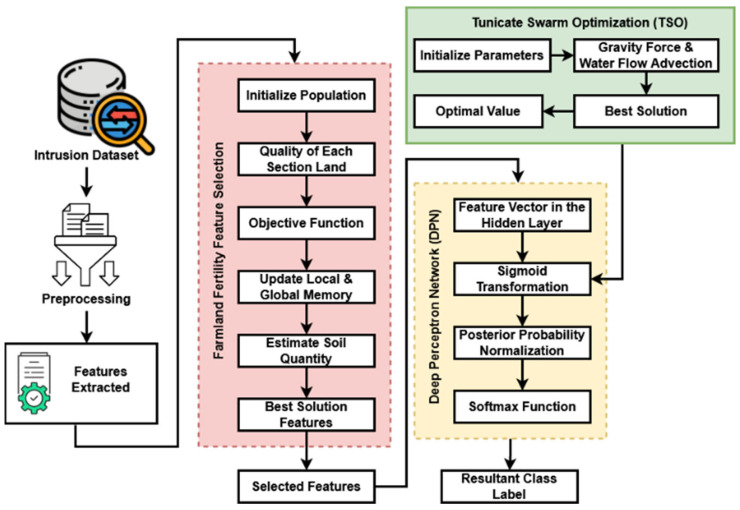
Workflow model.

**Figure 3 sensors-23-09294-f003:**
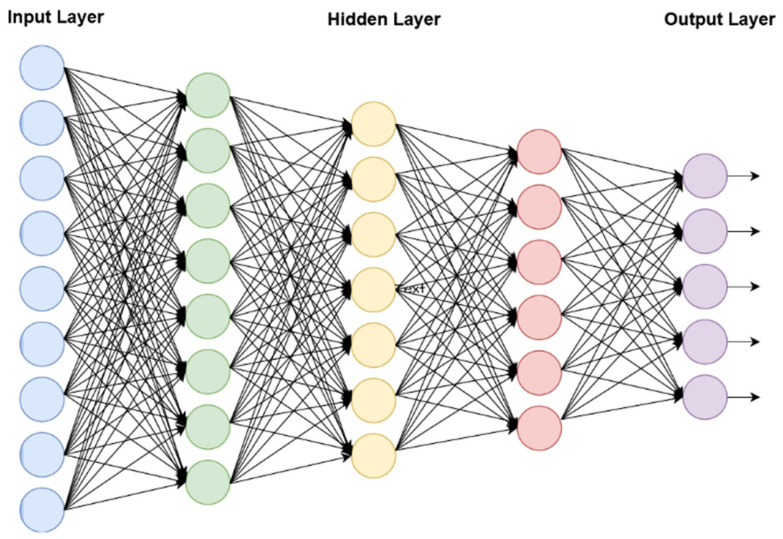
Structure of DPN.

**Figure 4 sensors-23-09294-f004:**
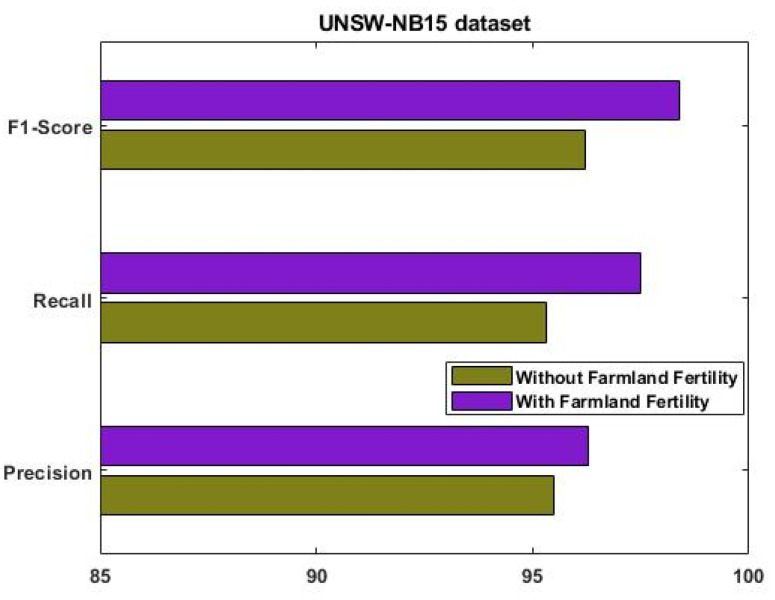
Performance analysis with and without optimization algorithms for IS-1.

**Figure 5 sensors-23-09294-f005:**
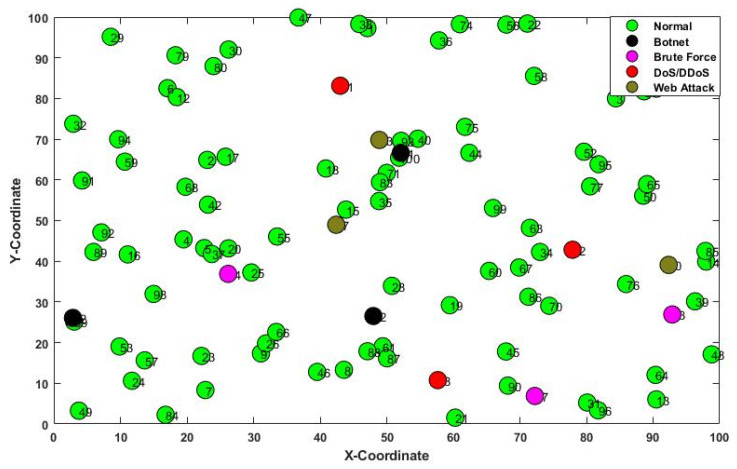
Network deployment.

**Figure 6 sensors-23-09294-f006:**
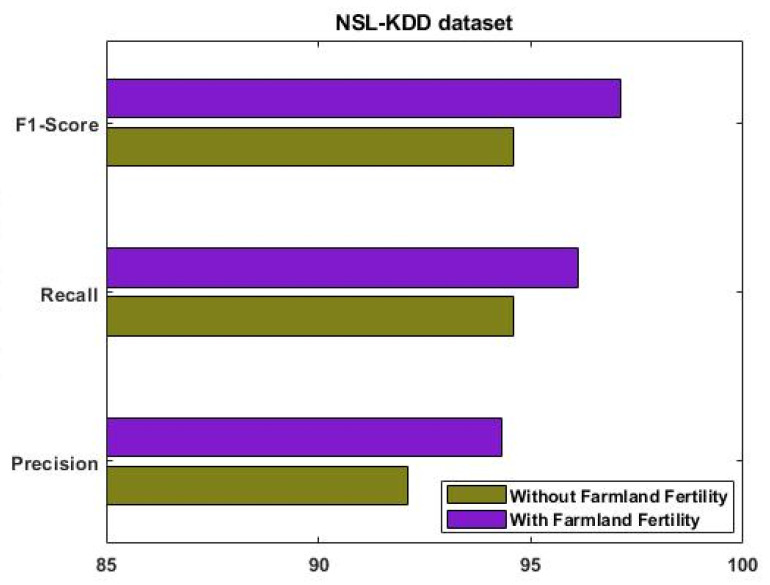
Performance analysis with and without optimization algorithms for IS-2.

**Figure 7 sensors-23-09294-f007:**
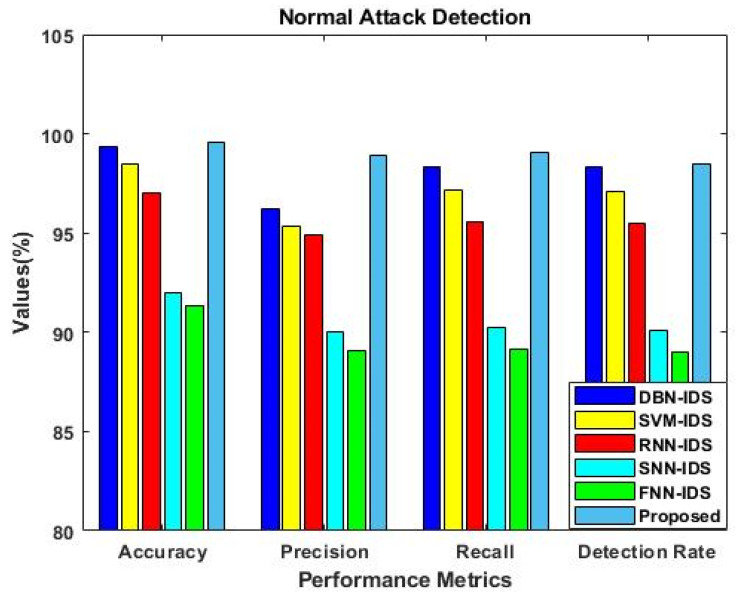
Comparative analysis with other IDS approaches for normal attacks.

**Figure 8 sensors-23-09294-f008:**
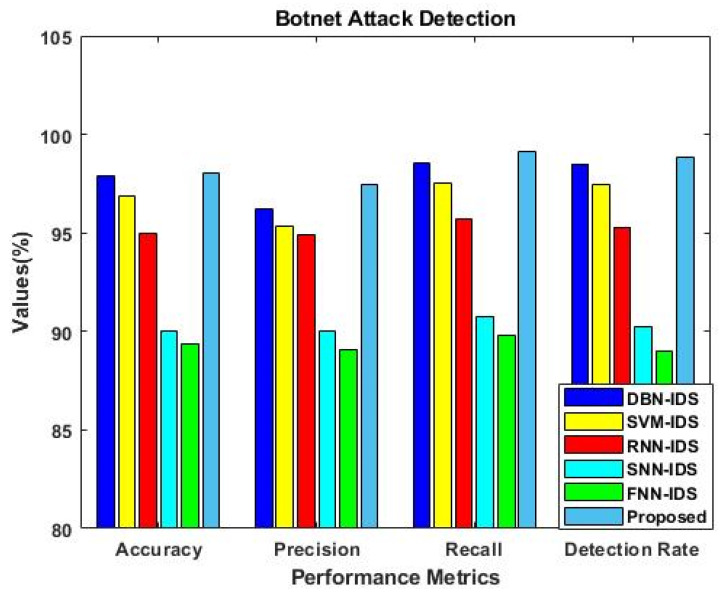
Comparative analysis with other IDS approaches for Botnet attacks.

**Figure 9 sensors-23-09294-f009:**
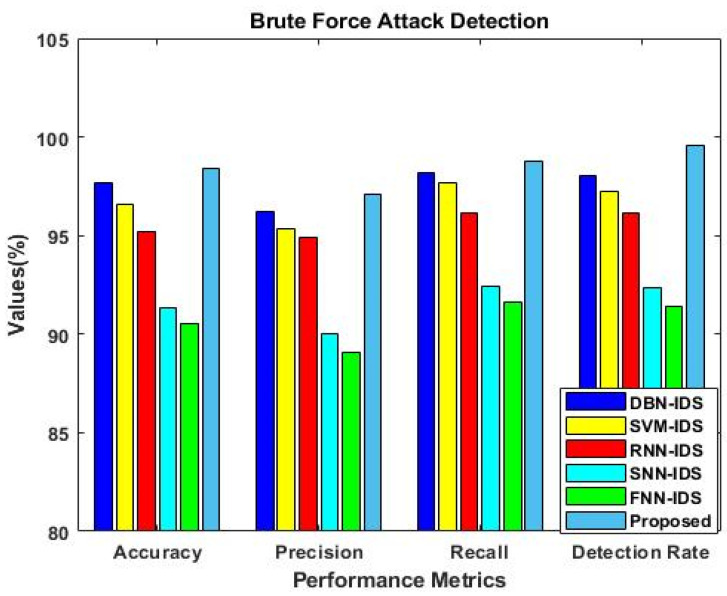
Comparative analysis with other IDS approaches for brute-force attacks.

**Figure 10 sensors-23-09294-f010:**
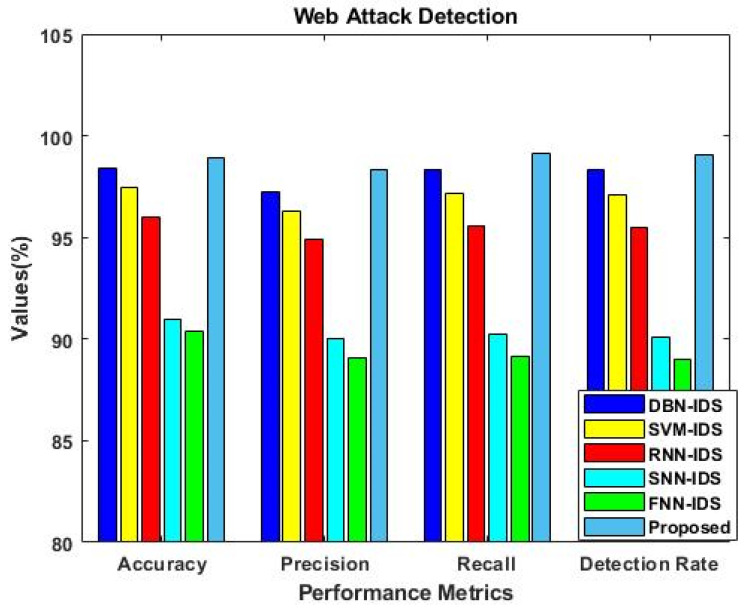
Comparative analysis with other IDS approaches for web attacks.

**Figure 11 sensors-23-09294-f011:**
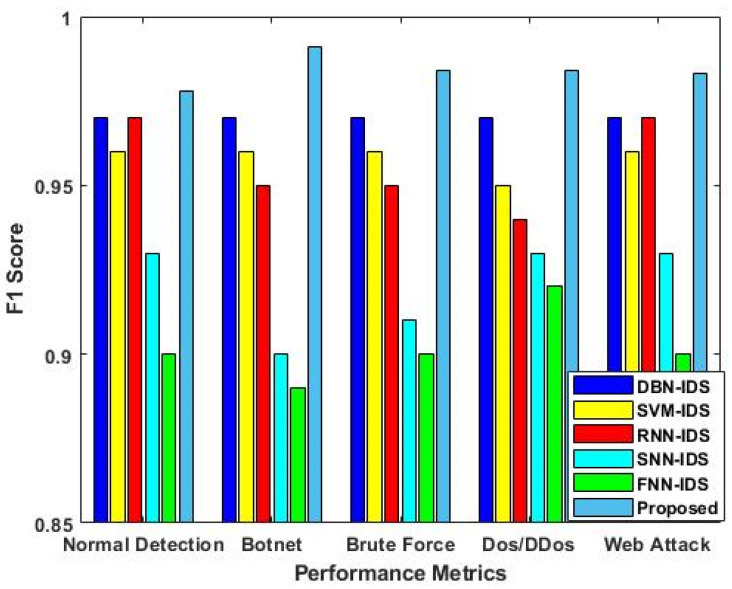
F1-score analysis with respect to different types of attacks.

**Figure 12 sensors-23-09294-f012:**
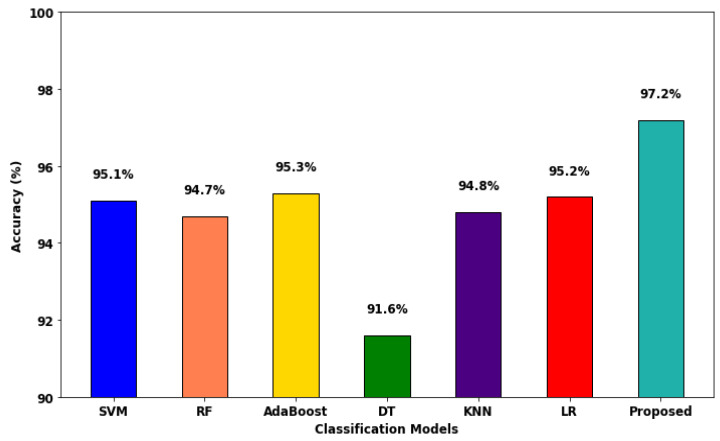
Comparative analysis with other IDS approaches based on accuracy.

**Figure 13 sensors-23-09294-f013:**
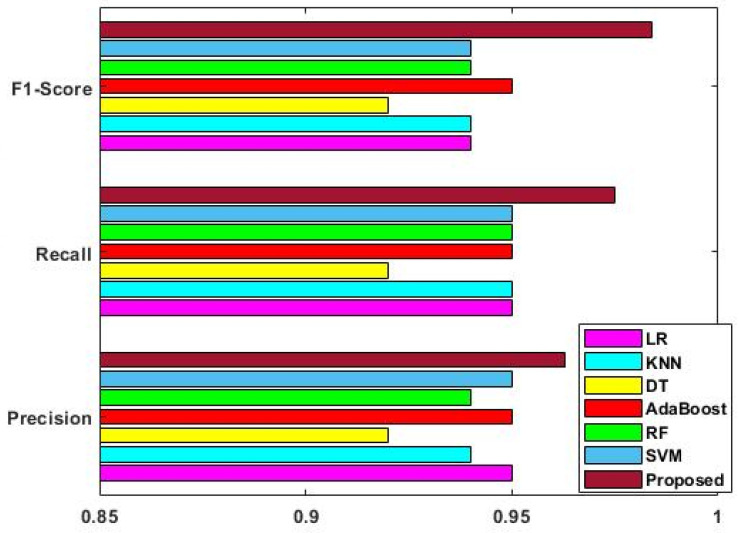
Intrusion detection performance analysis with other IDS approaches.

**Figure 14 sensors-23-09294-f014:**
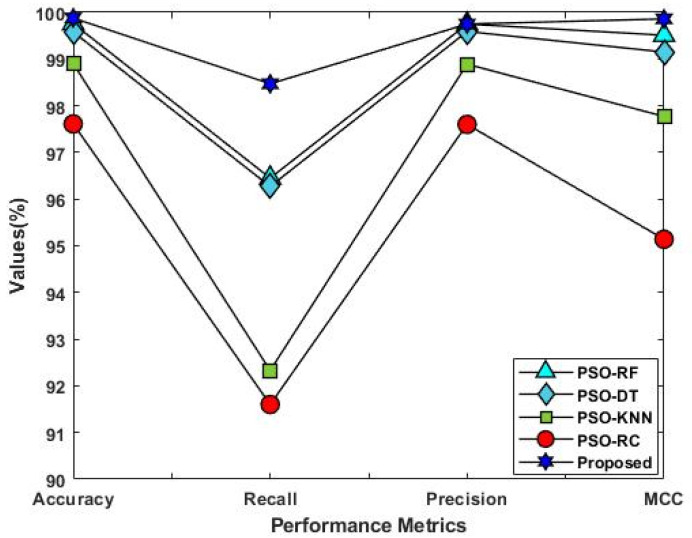
Overall comparative analysis with optimization-based intrusion detection approaches.

**Figure 15 sensors-23-09294-f015:**
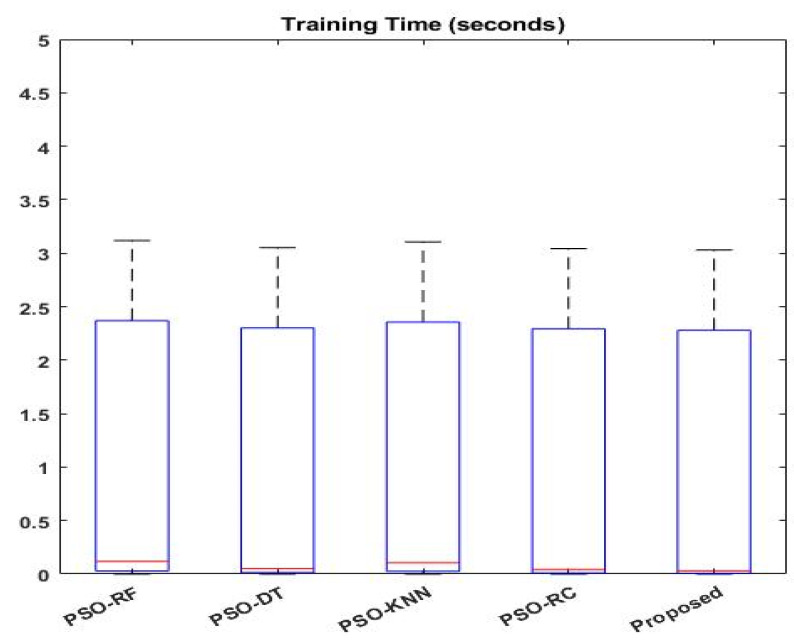
Comparative analysis with other IDS approaches based on training time (s).

**Figure 16 sensors-23-09294-f016:**
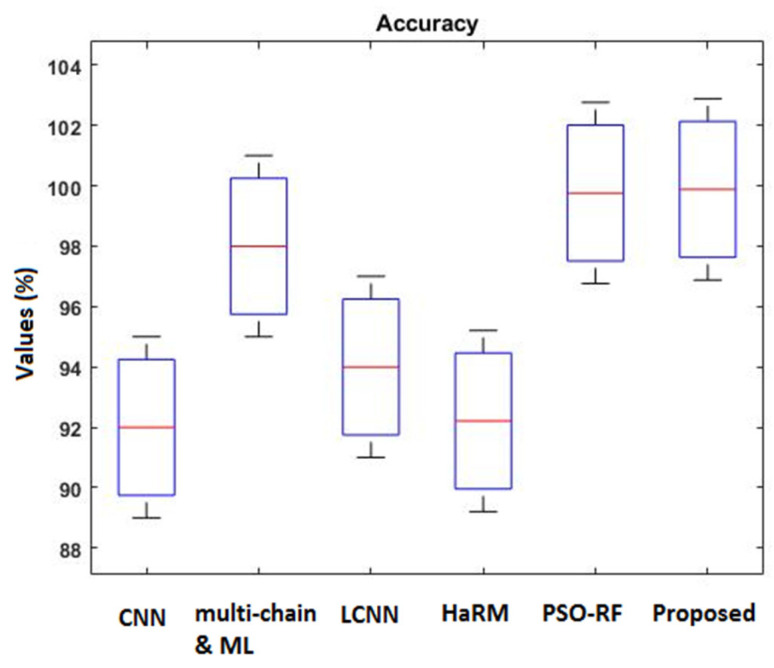
Comparative analysis based on accuracy.

**Figure 17 sensors-23-09294-f017:**
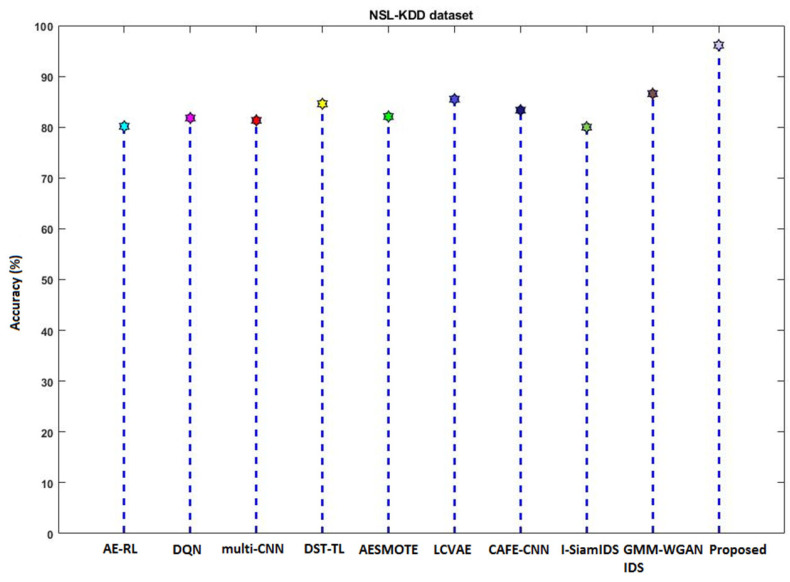
Accuracy analysis using IS-1.

**Figure 18 sensors-23-09294-f018:**
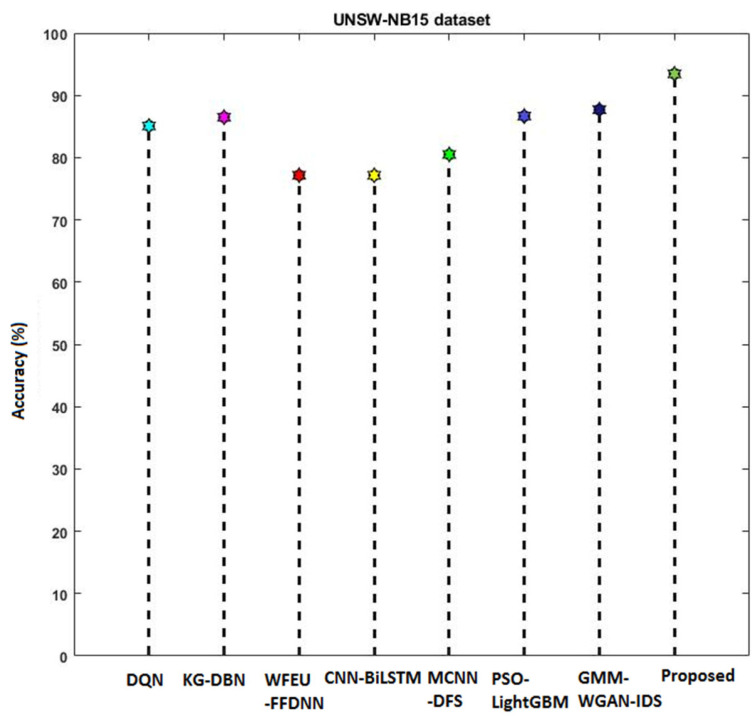
Accuracy analysis using IS-2.

**Table 1 sensors-23-09294-t001:** The table below provides a brief overview of the state of the art.

Reference	Methodology	Results	Limitations
Pundir et al. [24]	Investigated security challenges and requirements in WSN-IoT networks.	Identification of potential threats like eavesdropping, DoS, etc.	Low performance in various models.
Baraneetharan et al. [25]	Explored machine learning algorithms (classification, regression, clustering) for intrusion detection.	Comparative analysis based on prediction accuracy, energy, etc.	High false positive rate; increased time consumption for attack detection.
Jiang et al. [26]	Implemented a lightweight GBM-based cyber-physical system.	Enhanced smart-networking environment.	Low performance in various models.
Amouri et al. [27]	Cross-layered IDS framework using a linear regression model.	Detection of malicious activities like blackholes, DDoS, etc.	High false positive rate; increased time consumption for attack detection.
Singh et al. [29]	Comprehensive review of machine-learning-based intrusion detection approaches.	Highlighted strengths and weaknesses of various ML algorithms.	Low performance in various models; insufficient memory use during classification.
Damasevicius et al. [30]	Utilized LITNET-2020 dataset for classifying events; suggested other datasets.	Identification of normal and intrusive events in WSN-IoT systems.	Inability to handle massive datasets.
Safaldin et al. [31]	Binary grey-wolf optimization with SVM for intrusion detection, considering feature set reduction.	SVM with reduced feature set achieved efficient intrusion identification.	High curse of dimensionality.
Krishnan et al. [32]	Anomalous intrusion detection and prevention protocol for WSN-IoT.	Increased network reliability.	Excessive memory use during classification.
Jayanayudu et al. [33]	Hybrid SFL and ALO algorithms for an IDS framework; authors focused on energy efficiency with a greedy routing strategy.	Enhanced network efficiency; defence against fraudulent attacks.	Low performance in various models.

**Table 2 sensors-23-09294-t002:** Dataset details.

Attacking Classes	No of Samples
IS-1	
Normal	77,054
DoS	53,385
Probe	14,077
R2L	3749
U2R	252
UNSW-NB 15	
Normal	2,218,761
Generic	215,481
Exploits	44,525
Fuzzers	24,246
DoS	16,353
Reconnaissance	13,987
Analysis	2677
Backdoor	2329
Shellcode	1511
Worms	174

**Table 3 sensors-23-09294-t003:** Security analysis.

Methods	FAR	Accuracy	DR	No of Features	Time
Multi-agent IDS	L	L	VH	NA	NA
ARIMA-IDS	L	L	VH	NA	H
Lightweight IDS	VL	H	VH	NA	NA
Sensor IDS	L	H	VH	NA	NA
PSO-IDS	H	H	L	VH	NA
Evolutionary NN—MO IDS	VH	VH	VH	L	NA
GWO-SVM	VL	H	H	VL	VL
Proposed	VL	VH	VH	VL	VL

L—Low, VH—Very High, NA—Not applicable, H—High, and VL—Very Low.

## Data Availability

Data are contained within the article.

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
