# Peer review of "Development and Validation of a Cyber-Physical System Leveraging EFDPN for Enhanced WSN-IoT Network Security"

_sensors, 2023, doi:10.3390/s23229294_

Round 1

Reviewer 1 Report

Comments and Suggestions for Authors

In this paper, the authors claim to have developed a cyber-physical system based on the "Emphatic Farmland Fertility Integrated Deep Perceptron Network (EFDPN)" to enhance the security of the WSN-IoT. The authors have made a significant contribution in their paper, and the results of the proposed method also perform well in comparison of the other intrusion detection systems (IDS). However, there are major inconsistencies between the claims made and the actual contributions. I recommend that the authors conduct a thorough proofread of the paper with the guidance of cybersecurity experts. Some of my comments are as follows:

1. The main contribution of the authors is not clearly articulated in the abstract. While the authors state that they have developed a "cyber-physical system based on the Emphatic Farmland Fertility Integrated Deep Perceptron Network (EFDPN)," the contribution seems to involve the application of machine learning methods for classifying and identifying intrusions in cyber-physical systems. Clarification is needed.

2. The authors claim to have developed a cyber-physical system, but their contribution appears to be focused on an Intrusion Detection System (IDS). This inconsistency should be addressed.

3. In the abstract, the authors mention that "EFDPN is used to improve the security of WSN-IoT," while in lines 198-200, they state that "EFDPN is used to enhance the security framework of WSN-IoT." The authors should maintain consistency in their claims. Furthermore, the paper lacks a detailed discussion on how EFDPN is utilized to enhance security, making this a very general statement.

4. In line 218, the authors mention that "the proposed work aims to establish an efficient security framework for detecting intrusions in WSN-IoT systems." There appears to be an inconsistency regarding the primary objective of the authors in their development.

Author Response

Original Manuscript ID:  sensors-2651865       

Original Article Title: Development and Validation of a Cyber-Physical System Leveraging EFDPN for Enhanced WSN-IoT Network Security

To: Editor in Chief,

MDPI, Sensors

Re: Response to reviewers

Dear Editor,

Many thanks for insightful comments and suggestions of the referees. Thank you for allowing a resubmission of our manuscript, with an opportunity to address the reviewers’ comments.

We are uploading (a) our point-by-point response to the comments (below) (response to reviewers), (b) an updated manuscript with yellow highlighting indicating changes, and (c) a clean updated manuscript without highlights (PDF main document).

By following reviewers’ comments, we made substantial modifications in our paper to improve its clarity, English and readability. In our revised paper, we represent the improved manuscript such as:

(1) Revised Abstract, (2) Revised Introduction, (3) Results section, (4) Discussions and Conclusion sections.

We have made the following modifications as desired by the reviewers:

Best regards,

Corresponding Author,

Dr. Qaisar Abbas (On behalf of authors),

Professor.

Reviewer 2 Report

Comments and Suggestions for Authors

The work Introduces a new method and integrated it into practical applications, and validated its effectiveness through extensive experiments. The work lies in its innovative approach. It proposes a powerful cyber-physical system based on the EFDPN framework,  which primarily consists of the following three models:

1.Emphatic Farmland Fertility Integrated Deep Perceptron Network (EFDPN)

2.Farmland Fertility Feature Selection (F3S) technique

3.Deep Perceptron Network (DPN) classification algorithm

The following weakness should be improved:

1. NSL-KDD and UNSW-NB15 datasets are designed for intrusion detection in computer networks and are not specifically tailored for intrusion detection in WSN-IoT scenarios.

2. This article does not explore the computational overhead or resource requirements of the system, which may be key considerations for practical implementation.

3. The abstract contains a statement 'In the classification phase, it uses the Tunicate Swarm Optimization (TSO) model to improve the sigmoid transformation function. This makes predictions more accurate. '  Which should be clarified more clearly. 

4. A pie chart is suitable for displaying a whole divided into various parts, rather than for comparing the accuracy of independent methods. (Figure 12)

5. Figure 13 depicts traditional machine learning algorithms that are not comparable to neural networks

Author Response

(The authors gave the same response as above.)

Round 2

Reviewer 2 Report

Comments and Suggestions for Authors

All comments had been revised. 

Author Response

We have added high-quality figures as requested by editor

Thank you.